# The role of online social networks in improving health literacy and medication adherence among people living with HIV/AIDS in Iran: Development of a conceptual model

Azam Bazrafshani[1], Sirous Panahi[1]*, Hamid Sharifi[2‡], Effat Merghati-Khoei[3‡]

**1** Department of Medical Library and Information Science, School of Health Management and Information Sciences, Iran University of Medical Sciences, Tehran, Iran, **2** HIV/STI Surveillance Research Center, and WHO Collaborating Centre for HIV Surveillance, Institute for Futures Studies in Health, Kerman University of Medical Sciences, Kerman, Iran, **3** Iranian National Centre of Addiction Studies (INCAS), Institute of Risk Reduction, and Sexual & Family Health Division, Brain & Spinal Cord Injury Research Centre (BASIR), Institute of Neuroscience, Tehran University of Medical Sciences, Tehran, Iran

☯ These authors contributed equally to this work.
‡ HS and EMK also contributed equally to this work.
* Panahi.s@iums.ac.ir

**Data Availability Statement:** Due to ethical issues and institutional restrictions, we are not permitted to publish full-text of interviews with patients in

## Abstract

Online social networks have been used to enhance human immunodeficiency virus (HIV) and acquired immunodeficiency syndrome (AIDS) prevention, diagnosis, and treatment programs worldwide. This study aimed to develop a conceptual model of using online social networks in improving health literacy and medication adherence among people living with HIV/AIDS in Iran. This mixed-method sequential exploratory study was conducted in three phases. Firstly, a series of semi-structured interviews with a purposive sample of 29 HIV-positive patients were conducted to investigate the perceptions and experiences of HIV-positive patients about using online social networks to support health literacy and medication adherence. Thematic analysis was used to analyse qualitative interviews, extract potential components, and design a conceptual model. Then, a Delphi study with 27 HIV-positive patients was subsequently conducted to examine the consensus of patients on the proposed model. Finally, the trustworthiness and credibility of the proposed model were reviewed and evaluated by expert panel members from epidemiology and public health. Seven themes and 24 sub-themes emerged from the qualitative interviews. Five themes encompassed components of online social networks that supported communication and information-seeking behaviour of people living with HIV/AIDS. The two other themes encompassed social support and health-related outcomes including medication adherence. The credibility of the proposed conceptual model was confirmed methodologically using the expert panel and Delphi technique. Our findings highlighted that using online social networks has empowered Iranian people living with HIV/AIDS, making them more connected, safe, and able to access HIV/AIDS-related information and services. The role of online social networks in improving health literacy and medication adherence was also demonstrated in a conceptual model to understand the supportive components of online social

qualitative studies. The codes, themes, and sample testimonies from some interviews are provided as a supporting information file. In addition, the full-text of interviews with patients can be made available upon request to researchers who meet the criteria for access to confidential data. For more information, please send a request to the Vice-chancellor for Research, Iran University of Medical Sciences (research-m@iums.ac.ir).

**Funding:** This study was part of a PhD. dissertation supported by Iran University of Medical Sciences (No: IR.IUMS.REC 1396.9421623002). However, the authors received no specific funding for this study.

**Competing interests:** The authors have declared that no competing interests exist.

networks in the HIV care continuum as well as customized interventions to improve the success of antiretroviral therapies.

## Introduction

According to the Joint United Nations Programme on HIV/AIDS (UNAIDS) data, approximately 59,000 people were living with HIV in Iran at the end of 2019 [1]. Injection drug use is known as the primary source of HIV transmission in Iran. With an estimated prevalence of 15.2%-13.8% in 2010 and 2013, people who inject drugs (PWID) continue to be among the most vulnerable populations to acquire HIV infection [2]. However, emerging evidence indicates an increasing pattern of HIV incidence among women and female sex workers. Therefore, female sex workers are considered the second most at risk population for HIV transmission [2–4]. Prostitution and sex work are extremely illegal and stigmatized in Iran due to conservative cultural norms regarding sexual habits. Illicit drug use and illegitimate sexual practices have made HIV/AIDS a neglected and taboo subject over the years and consequently have posed major challenges to successful treatment in Iran.

Retention in care and adherence to antiretroviral therapy (ART) regimens are crucial for successful HIV treatment. Yet in Iran, only 20% of Iranian people living with HIV/AIDS (PLWHA)are engaged in medical care for treatment and only 17% are ART adherent and virally suppressed [5, 6]. Additionally, limited access to ART, poor medication adherence, and limited access to key populations (men who have sex with men, PWID, people in prisons, female sex workers and their clients, and transgender people) are threatening the success of HIV prevention and control programs in the country [7, 8].

In the light of these social conditions, Internet and online social networking (OSN) tools have emerged as promising sources of delivering health services, health information, and social support for PLWHA and their healthcare providers [9–11]. These technologies globally provide advantages for PLWHA to cope with the complicated nature of HIV through communication with peers and care providers, access to and receiving health information, share health and health care data including personal experiences and social support [12–14]. OSN tools have the potential to empower PLWHA, not only by producing patient-centered healthcare services but also by challenging the stigma and discrimination that PLWHA are likely to encounter or affected by.

A review of the existing literature has demonstrated that hundreds of thousands of PLWHA across the world, sharing their concerns about the disease, have taken advantage of OSN applications and online support groups to exchange resources and support. The precursor literature review helped set the diverse spectrum of OSN tools, the current global application, and the range of interventions designed by healthcare providers to address the challenges of the HIV care continuum. The review of the existing literature also demonstrated that the current and emerging applications of OSN tools are extensive, and require complex global movements that transcend nationalities, cultures, sexual behaviours, community and healthcare services, and patient outcomes. However, exploration and review of the literature to identify any existing evidence on the use of OSN tools by the Iranian PLWHA or healthcare providers do not return any documents or no answers. In addition, no existing conceptual model or theoretical framework was found to ensure the efficacy of OSN tools as HIV/AIDS prevention and control strategies in Iran. In addition to the fact that 94.4% of the Iranian population own a mobile phone [15] and 69.1% have access to the Internet [16] with 40 million

active social media users [17], OSN platforms are considered a powerful and indispensable venue in promoting HIV prevention and control programs in Iran. Although positive evidence of using OSN technologies on health communication, empowering process, and patient engagement has been well reported, there has been little attention devoted to understanding the mechanisms through which participation in OSN platforms might be related to treatment outcomes including ART adherence in patients with HIV/AIDS. Therefore, this study aimed to develop a conceptual model of using online social networks in improving health literacy and medication adherence among people living with HIV/AIDS in Iran.

## Methods

Iran University of Medical Sciences research ethics committee approved this study (No: IR. IUMS.REC 1396.9421623002). Written consent was obtained from participants before conducting interviews.

To achieve the study goals, a sequential mixed-method exploratory study was conducted in three phases.

Phase 1: semi-structured interviews were done to propose the preliminary list of OSN roles in improving health literacy and medication adherence among PLWHA

Phase 2: A Delphi study was undertaken to determine the consensus of PLWHA on the proposed model

Phase 3: An expert panel was done to evaluate the validity, credibility, and transformability of the model

### Phase 1: Semi-structured interviews

This phase involved semi-structured interviews with a sample of 29 Iranian PLWHA to draw up an academic basis of concepts and dimensions of using OSN tools and determining the key roles of OSN tools and the factors that drive health literacy and medication adherence.

The inclusion criteria were 1) participants over 18 years of age, 2) use of HIV/AIDS OSN for at least six months 3) could speak and understand the Farsi language, and 4) willingness to participate in the interview. A purposive sample of 20 PLWHA eligible for the study was recruited from two dominant public health centers (Imam Khomeini Hospital, and Zam Zam Clinic) in Tehran. The remaining 9 study samples were recruited from centers distributed across the country through the snowball sampling method. These centers compromised the largest proportion of target populations and most PLWHA refer to these centers for their treatment and care.

Interviews were conducted face to face (n = 21) for those participants who were in Tehran. For the participants living in other areas of the country, we contacted them via telephone (n = 8). An interview guide was developed by a review of relevant literature. This interview guide focused on how PLWHA used OSN and their opinions about the roles of OSN for improving health literacy and medication adherence. The interview guide was primarily assessed by two expert reviewers. It was subsequently pre-tested with three target population members before the implementation.

The interviews followed a semi-structured design. However, the order of the questions and answers could vary according to the responses of the participants. The objectives and the activities that were involved in the study were explained to the participants. The investigator's contact details were provided and participants' confidentiality was assured. An experienced interviewer with a background in qualitative research and well-skilled in interviewing patients

conducted the interviews in the Farsi language. Interviews were audio-recorded and transcribed verbatim. Interviews ranged between 14–90 minutes in length (Mean = 43 minutes) and the longest interview lasted approximately 90 minutes. Interviews were conducted until the researcher realized that data saturation had been attained and no new information was forthcoming.

Braun and Clarke's framework for content analysis was used in data analysis, in which codes and themes were derived inductively from the interview transcripts and notes. Manual data analysis by using MAXQDA 12 (VERBI GmbH, USA) was started after the first three interviews and conducted concurrently with data collection. Initial codes were generated by labelling data extracts with a code reflecting the meaning of the extract (S1 Appendix). These codes were then collated into potential themes. This coding was followed by an iterative process, in which the themes were reviewed with the coded extracts and the entire dataset. This review was conducted to ensure the consistent application of codes and themes, and clear definitions and names for each theme.

According to the interviews and extracted themes, a preliminary list of OSN roles was completed (S2 Appendix). This list comprised 24 dimensions divided into seven domains. This list served as a basis for the first round of the Delphi process.

## Phase 2: The Delphi study

In this phase, the Delphi technique was used to obtain a consensus about the critical roles of OSN technologies in improving health literacy and medication adherence.

The Delphi technique comprised the following steps [18, 19]:

- Identifying and selecting potential experts based on predefined criteria

- Developing a questionnaire and transmitting the questionnaire to panel members in different rounds

- Collecting and analyzing round responses

- Developing feedback for panelists for subsequent rounds

- Evaluating consensus and reporting results

The Delphi technique was used because it is useful for consensus-building where information gathering and feedback from stakeholders (PLWHA) is difficult due to geographical barriers and the need to guarantee anonymity. Delphi method is also useful for developing new concepts and setting the direction of future-oriented research including conceptual models and frameworks [20, 21]. In our research, we strove for a consistent model of the critical roles of OSN tools in improving health literacy and medication adherence among PLWHA to induce consensus.

A unique feature of the Delphi method is that decisions are made through informed consensus of knowledgeable participants. Therefore, selecting a qualified Delphi participant is extremely critical [20]. The size of the Delphi panel is rather associated with the group dynamic in obtaining consensus than the statistical power [22]. Therefore, small groups of participants may be considered sufficient in some research areas [23]. In this study, participants were recruited through purposive and snowball sampling from PLWHA who consistently used OSN technologies in recent months. An invitation letter explaining the study and inviting them to participate in the online survey was distributed through some popular and well-recognized OSN groups. The investigator's contact details were provided and the participants' confidentiality was assured in the invitation letter. A total of 127 PLWHA who were using OSN

technologies were invited to participate in this study. Finally, 26 PLWHA accepted the invitation and participated in the Delphi study (15 women, 11 men).

The Delphi study comprised 1 round. In this round, participants were required to independently rank the importance of a total of 24 statements across seven domains using a five-point Likert scale (ranging from completely disagree to completely agree). Statements for the survey were developed from the qualitative interviews primarily conducted with 29 PLWHA, the research team's expertise, and a review of existing literature. To meet the study objectives, the survey was divided into three sections: demographic data of respondents (age, gender, and education), statements about health literacy, and medication adherence among PLWHA. One author (AB) independently analyzed the responses and calculated the consensus level.

The percentage of total score, median and inter-quartile range for each question was calculated.

The consensus level for this phase was determined as at least 75% and items with higher than 75% consensus were accepted. Descriptive statistics and frequency tables were used to analyse and summarise data in this phase. Since the favourable consensus level on all statements was obtained in the first round of the Delphi study, a further survey round was not required.

## Phase 3: Nominal Group Technique (NGT)

NGT and Delphi studies are the most common consensus methods used in health research for prioritizing ideas, problem-solving, or idea generation [24]. The Delphi method usually uses multi-stage self-completed surveys with individual feedbacks to determine the informed consensus from a larger group of knowledgeable participants. However, The NGT uses face-to-face discussions in small groups to explore stakeholders' and consumers' perceptions. The Delphi surveys are often posted or e-mailed to participants, but the NGT requires participants to attend a meeting. The favourable size of an NGT group ranges between five and seven members [25].

In this phase, a group of five participants was purposively selected to verify and prioritize key roles of OSN tools and evaluate the credibility and transformability of the proposed conceptual model. The panel included key researchers and health professionals with practical experience in either HIV/AIDS programs or decision-making in national public health settings. Inclusion criteria for selecting expert panel members were 1) at least 3 years of working experience in clinical (general practitioner) or academic settings (researcher), and 2) willingness to participate in the study. Several extra criteria were further added for choosing policymakers (having at least 3 years of practical experience in either HIV/AIDS care programs or decision-making in national public health settings) and for choosing stakeholders (Having participated in any HIV/AIDS online social networks; having HIV for at least 5 months).

Potential panel members (n = 5) were identified through their clinical leadership roles and with the assistance of the HIV/AIDS Office, the Ministry of Health provides HIV/AIDS care services.

The NGT phase comprised the following stages:

- Silent generation of ideas (brainstorming)

- Round-robin (recording of comments and ideas)

- Clarification (discussion on the proposed list of OSN roles)

- Voting (ranking)

The NGT members were primarily approached by email with a short statement of the purpose of the meeting. The conceptual model initiated from the qualitative interviews and the

Delphi study with PLWHA was subsequently discussed in the NGT panel to evaluate the trust-worthiness and identify whether the conceptual model adequately reflected relevant factors representing OSN roles in improving health literacy and medication adherence in PLWHA. Based on this step, several modifications were added to the conceptual model, but no factor was excluded. The consensus level for the phase was determined as at least 75% agreement.

## Results

### Phase I: Semi-structured interviews

Participants mostly were PLWHA aged 31–40 years (n = 17, 58.6%), and men (n = 18, 62.1%). Their level of education was high school or lower (n = 26, 89.6%), and did not report a history of drug injection or addiction (58.6%, n = 17). Also, about half of the respondents were unemployed (48.3%, n = 14). The main source of HIV transmission was sexual behaviour (51.7%, n = 15) and needle or syringe use (20.7%, n = 6). About half of the respondents were diagnosed with HIV for over10 years (n = 13, 44.8%). (Table 1).

Participants appeared on average 3.5±5.7 hours per day online. Telegram (n = 24; 83%) and WhatsApp (n = 6; 21%) were the most attracted OSN platforms among the participants. Few participants also reported using Facebook (n = 2; 7%) for communication with physicians and peers from other countries. Over half of the PLWHA (n = 21, 72%) who participated in the study had frequently used pre-existing platforms (groups) established by health care providers or community-based organizations. Thirteen (44.8%) participants reported starting their groups for communicating with peers.

The themes that emerged from the semi-structured interviews with PLWHA are shown in Table 2. According to this table, the role of OSN technologies as a support mechanism in improving health literacy and medication adherence of PLWHA could be described through a set of themes and sub-themes. The "health literacy" category encompassed the supportive properties of OSN platforms that improve the health literacy of PLWHA with five themes and 19 subthemes including communication support, access to information, sharing personal experiences/ success stories, increasing knowledge and awareness about HIV, and perception and enhancing attitudes towards HIV. The "medication adherence" category encompassed the supportive properties of OSN platforms that affect the adherence of PLWHA to anti-retroviral medications with two themes and five sub-themes including social support and health-related outcomes.

**Theme 1: Communication support.** *Improving patient-provider communication*. Findings from semi-structured interviews suggest that OSN platforms facilitated social interactions with physicians and care providers (n = 17, 58.6%). A variety of health professionals have participated in OSN platforms to communicate with PLWHA including general practitioners, midwives, clinical psychologists, infectious disease specialists, and social workers. According to the respondents' perspectives, OSN platforms have changed patient-physician relationships and provided the PLWHA with the advantage of reduced unnecessary use of healthcare facilities and consequently the cost of healthcare services. However, the participation of some physicians and psychologists in OSN groups was influenced by professional activities and social responsibilities, and some of these users could not be active in OSN groups consistently.

*Improving peer communication*. It was often reported that OSN platforms enabled Iranian PLWHA to stay easily connected with friends and peer groups. Many participants (n = 19, 65.5%) found it easier to communicate with their close friends through OSN groups and become aware of their routine life, work, feelings, and emotions, and arrange off-line activities. In addition, they could refer to their friends and peers for immediate questions, feedback, and assistance.

**Table 1. Demographic characteristics of a qualitative sample of PLWH in Iran.**

| Demographic and behavioural characteristics | Frequency (%) |
|---|---|
| Recruitment site | |
| Imam Khomeini Hospital | 21 (72.4) |
| Zam Zam Clinic | 8 (27.6) |
| Residence at the time of interviews | |
| Tehran | 20 (69.0) |
| Ahwaz | 4 (14.0) |
| Karaj | 1 (3.4) |
| Bushehr | 1 (3.4) |
| Bandar-Abbas | 1 (3.4) |
| Qum | 1 (3.4) |
| Kerman | 1 (3.4) |
| Age | |
| 18–20 | 0 (0.0) |
| 21–30 | 2 (6.9) |
| 31–40 | 17 (58.6) |
| >40 | 10 (34.5) |
| Gender | |
| Men | 18 (62.1) |
| Women | 11 (37.9) |
| Education | |
| High school or below | 25 (86.3) |
| Some college† | 1 (3.4) |
| College or above | 3 (10.3) |
| Employment status | |
| Employed | 12 (41.4) |
| Disabled/Retired | 3 (10.3) |
| Unemployed | 14 (48.3) |
| History of drug injection or abuse | |
| Yes | 12 (41.4) |
| No | 17 (58.6) |
| Source of transmission | |
| Sexual behavior | 15 (51.7) |
| Needle or syringe use | 6 (20.7) |
| Blood | 3 (10.3) |
| Unknown | 5 (17.3) |
| Years of living with HIV/AIDS | |
| ≤5 years | 10 (34.5) |
| 6–10 years | 6 (20.7) |
| >10 years | 13 (44.8) |

† Includes current students

*Building trust among users*. Online social interactions and communication with peer groups and health care providers were significantly associated with building and maintaining trust among OSN users (n = 27, 93.1%). According to semi-structured interviews, Iranian PLWHA tend to trust physicians and health workers by following their advice and treatment instructions. PLWHA also tend to trust other group members by sharing their feelings, thoughts,

**Table 2. Themes and subthemes related to the role of online social networks in improving health literacy and medication adherence of PLWHA.**

| Themes | Sub-theme | Example quotes |
|---|---|---|
| Communication support | Improving patient-provider communication | "In this online community, there is a variety of medical specialties including social workers, internal physicians, midwives, and psychologists sharing information with HIV positive patients and answer patients' questions daily" [Man, 41 years old] |
| | Improving peer communication | "We have been friends for 5 years . . . Here we contact each other in the group for the first time and become very close friends . . . We are very interdependent and always talk about our smallest to our biggest problems here in the group" [Woman, 35 years old] |
| | Building trust among users | "Personally, if someone gives me advice about the disease, I do not accept it in the first place. I first try to evaluate whether the source of information is reliable or not . . .once the quality and credibility of the information is confirmed, I would accept it" [Man, 39 years old] |
| | Creating collective identity, a sense of belonging and attachment among users | "It's very valuable to me to see that there is a connection between the patient and that they feel a responsibility to relieve their pain while no one knows where you are from . . .we are like a family" [Woman, 37 years old] |
| Access to information | Enabling remote access to information | "During this period, I think [that] it has helped a lot to those [who are] not able to visit counselling centers or could not remember their questions at the visit time. Since there are some medical staff and physicians in the group; we can ask our questions 24/7. This could particularly help members from other cities to access health professionals and physicians online" [woman, 45 years old] |
| | Maintaining confidentiality and anonymity | "Most members deleted their profile pictures or use anonymous Ids so that nobody can recognize them as HIV positive members" [Man, 39 years old] |
| | Improving the hidden population's access to health information | "Many [homeless] guys in Ahwaz rely on me to ask me their questions . . .I share their questions with doctors in the group and then send the doctor's advice to my friends reciprocal" [Man, 41 years old] |
| Sharing personal experiences/ success stories | Offering interactive Q and A for patients | "At the time I was added to the group, I knew nothing about HIV. By reading other members' posts and questions/answers shared by physicians, I became aware of complications of the disease and got informed how to deal with them" [Man, 40 years old] |
| | Improving users' motivations to share personal experiences | "During online interactions among patients, one learns how to express his HIV status instead of hiding it . . . By presenting ourselves to a group of people without any discrimination or stigma, we are motivated to talk about HIV even among our family or other people" [Man, 38 years old] |
| | Enhancing observational learning | "In this group, we have different kinds of HIV patients. Some are newly diagnosed patients. Some have been living with HIV for a long time. Some have dealt with the disease successfully but some members are experiencing many challenges. By observing all these patients, one can learn how to manage his situation" [Woman, 39 years old] |
| | Empowering patients in making a decision | "Communication with other patients in the group helped me identify best practices among my peers and apply them based on my situation." [Man, 35 years old] |
| | Creating and maintaining social capital in online communities | "Some members in the group are extremely active, share their personal experiences without any fear . . . They can affect other members by their thoughts, behaviours and speeches" [Man, 39 years old] |
| Increasing knowledge & awareness about HIV | Improving knowledge about symptoms and clinical manifestations of HIV infection | "I'd always thought that an HIV-positive person has a strange appearance with horrible complications. He is very thin or dying . . . If I knew that an HIV-positive patient might have no symptoms, then I could communicate with him/her with caution" [Woman, 38 years old] |
| | Improving knowledge about drug benefits/side effects | "I used to be in some patient groups and I didn't know much about HIV, but now I'm in this group . . . It is much better . . . I have learned many new things about HIV including nutrition, drug side effects, herbal drugs, and treatment regimens . . . [I] asked all my questions or read other members' questions about HIV . . . this has increased my knowledge extensively" [Man, 25 years old] |
| | Providing information about available health services and service providers | "By joining this group and communicating with other patients, I got familiar with health centers near my location that provide services to HIV-positive patients free of charge and deliver health packages for at-risk women and sex-workers" [Woman, 31 years old] |
| Enhancing perception &attitudes towards HIV | Enhancing the perception of users about acquiring HIV infection | "Acquiring HIV infection is unbelievable to most people. I denied my HIV-positive test result for months and couldn't believe that I'd got an HIV infection. By getting involved with other HIV-positive patients, I've recognized that the risk of acquiring HIV treats all people irrespective of their sex or socioeconomic class." [Woman, 32 years old] |
| | Enhancing perception about negative outcomes and severity of the disease | "Most people have misconceptions about HIV/AIDS. They often think that once a person is infected with HIV, he will die in a few years or his mouth will be crooked (for example) or he can no longer get married or have a healthy child" [Woman, 40 years old] |
| | Enhancing the perception of users about the potential benefits of adapting prevention behaviours | "I'd always thought that HIV-positive patients are not required to use a condom while having sex with each other, but the doctor in the group convinced me that we must use preventive strategies once communicating with other patients rather than healthy people" [Man, 40 years old] |
| | Improving users' knowledge about social, personal, environmental, or economic challenges of living with HIV infection | "Losing a job, getting divorced from an HIV-negative wife, and leaving home are frequently reported by HIV-positive members in online groups. Most of my friends experienced such problems after being diagnosed with HIV infection" [Woman, 56 years old] |

*(Note: "Health literacy" spans the Sharing personal experiences, Increasing knowledge & awareness, and Enhancing perception themes)*

*(Continued)*

**Table 2.** (Continued)

| Themes | | Sub-theme | Example quotes |
|---|---|---|---|
| Medication adherence | Social support | Facilitating emotional support exchange among users | "Sometimes you need someone thinking about you or you need to feel valuable when you post in a group and then some users like or comment on your post, this simply makes you feel better. When you are in a bad mood or feel depressed, there is someone to talk with you, this makes you feel you are not alone at all. Talking about your disease in a group of peers makes you feel self-confident and powerful" [man, 40 years old] |
| | | Facilitating instrumental support among users | "In the Kermanshah earthquake, our friends needed help, we collected a lot of help from the group members and sent them . . . Someone in Kermanshah wanted to commit suicide. We went to Kermanshah and tried to convince her. Now she is an artist and works in the positive club voluntarily" [Man, 35 years old] |
| | Health-related outcomes | Boosting motivation and self-confidence | "Patients are constantly following other users' health status and their personal behaviours . . . They could be inspired and motivated by the success stories of other users and learn from their experiences . . . Looking at patients who successfully managed the HIV complications improves the motivation and confidence of recently diagnosed patients and encourages them to take an active role in their healthcare management" [Woman, 35 years old] |
| | | Improving self-efficacy and self-care behaviours | "While chatting with online patients and looking at their very positive attitude, I'd always felt a twinge of envy for them and wished to be like them . . . This strong feeling reinforced me to stop drug use and start my ART regimen" [Man, 37 years old] |
| | | Improving medication adherence | "Online communication with other patients in this group offers the opportunity of getting up-to-date information about the disease and becoming aware that what would happen to them if they did not adhere to their medications correctly or interrupted the treatment . . . It could help non-adherent patients stick to their regimen permanently" [Man, 47 years old] |

opinions, experiences, images, and files. However, the anonymity of users, sharing false or misleading information, and poor engagement with other users were perceived as barriers to emerging trust among OSN group members. In addition, the belief of PLWHA about specific characteristics of other users including their competence, ability, integrity, honesty, and benevolence, influenced their trusting intentions and behaviours. For example, PLWHA tend to trust those community members with healthy behaviours, satisfactory clinical outcomes, and acceptable medication adherence.

*Creating collective identity, a sense of belonging and attachment among users*. OSN platforms created an environment where PLWHA could share their identity, personal experiences, and opinions without concerns about privacy and fear of being judged. In this context, PLWHA developed a sense of belonging to a collective identity (like a family) that led to their involvement and commitment to the group's norms and activities. Communication with HIV peer groups strengthened the sense of belonging and prevented patients from treatment interruption, social isolation, and feeling lonely. It seemed that a sense of belonging with online communities was associated with increased life satisfaction among PLWHA and made their presence in OSN groups enjoyable. Constant checking group messages, participating in online discussions, and sharing personal thoughts, opinions, and feelings with other users were stated by respondents as examples of attachment to the group activities.

**Theme 2: Access to information.** *Enabling remote access to information*. Remote access to health information or physician's advice was also identified as a notable benefit by some respondents (n = 10, 34.5%). As a matter of fact, OSN platforms have provided opportunities to disseminate health information, and personal experiences to distant members. The participants stated that OSN platforms enabled them to gain rapid access to reliable health information, and medical consultation services 24/7. The use of OSN platforms in emergency situations where it was not possible to visit a doctor or health centers, was also reported by some PLWHA participants in this study. According to the interviews, OSN platforms facilitated access of PLWHA from other geographic areas and distant locations across the country to updated information about the HIV care continuum.

*Maintaining confidentiality and anonymity*. Although respondents did not specify whether their use of OSN platforms was anonymous or required disclosure of identity, some of them

reported the importance of anonymity in OSN's use to communicate about HIV mostly due to the stigma associated with HIV (n = 4, 14.8%). Stigma was considered one of the most important barriers that prevented the PLWHA from accessing in-person HIV prevention and treatment programs. Therefore, using OSN platforms with anonymity and hidden social identity benefited the PLWHA to access health information and communicate with other members without the fear of discrimination and stigma. Combating HIV/AIDS stigmatization through engagement in online discussions about HIV and group activities was also reported by some PLWHA as a particular benefit of using OSN platforms.

*Improving the hidden population's access to health information.* It was reported that OSN platforms did particularly benefit hidden or hard-to-reach populations such as injecting drug users, and sex workers to access health information on OSNs easily (n = 5, 17.2%). According to the respondents' perspectives, due to high levels of stigmatization, hidden populations tend to avoid using community health services and HIV prevention programs. In addition, lack of access to mobile phones or the internet was reported as one of the most important barriers that prevented these populations from using OSN platforms. However, some participants reported that they disseminated new knowledge they acquired from health professionals to their PLWHA friends who did not use mobile phones or OSN platforms.

**Theme 3: Sharing personal experiences/ success stories.**   Sharing personal experiences were reported as one of the main supportive components of OSNs for Iranian PLWHA (n = 26, 89.6%). PLWHA often began searching for information about HIV/AIDS by consulting patients with similar HIV experiences. Respondents believed that online interactions with other similar patients and sharing lived experiences were effective strategies for learning how to cope with the disease.

*Offering interactive Q and A for patients.* Respondents (n = 24, 82.7%) believed that interactive Q and A in OSN platforms enabled them to share their lived experiences with other similar patients and helped them learn how to cope with the social, physical, and health consequences of the disease. In addition, participating in online discussions and reading about other patients' experiences helped them deal with challenges and problems encountered in later life. Three types of interactive Q and A services are usually provided for PLWHA in OSN platforms:1) medical consultations by physicians and health care providers,2) social Q and A services by social workers or NGOs, and3) peer consultations by community members.

*Improving users' motivations to share personal experiences.* Some respondents (n = 23, 79.3%) argued that reading about peers' experiences and communication with care providers motivated them to actively engage with online discussions and share their personal stories without the fear of stigma and judgment. Self-disclosure of HIV status, and communicating potentially stigmatizing information that had previously been kept hidden was reported as one of the most important benefits of OSN platforms for some PLWHA. Participants in this study also pointed out that communicants with peers and care providers encouraged them to actively discuss HIV and even disclose their HIV status with HIV seronegative partners or their family members voluntarily.

*Enhancing observational learning.* Some respondents (n = 19, 65.5%) mentioned that OSN platforms benefited them respond to their feeling of uncertainty and anxiety by observing and comparing their peers' behaviours, reading about their experiences, and comparing them to their own situations. Two types of observational learning activities were frequently reported by Iranian PLWHA: (1) seeking information about peers' experiences, thoughts, and feelings and (2) communicating with peers indicating similar experiences. It was often reported that PLWHA usually tried to perceive peers that had successfully managed their illness and controlled the negative consequences of HIV infection as role models and followed their experiences and actions to manage their own situation. In this context, PLWHA used social

comparison activities to assess their own situation, current reactions, and coping skills. Social comparison helped PLWHA feel hope or inspired them to improve their situations.

*Empowering patients in making decisions*. OSN platforms benefited PLWHA by empowering them in making treatment decisions (n = 14, 48.3%). Dissemination of experiential information among peer groups helped PLWHA assess the benefits and side effects of treatments, and identify best practices, and effective treatments.

*Creating and maintaining social capital in online communities*. Some respondents (n = 4, 13.8%) suggested that OSN platforms created and maintained social capital through online social interactions and social ties among users, sharing information, social norms, values, and trust and developing a sense of collective identity among users. Cooperation among PLWHA to achieve a common goal of protecting each other against negative consequences of HIV infection and improving quality of life made these online communities a source of cognitive development and personal growth.

**Theme 4: Increasing knowledge and awareness about HIV.** *Improving knowledge about symptoms and clinical manifestations of HIV infection*. Getting information about symptoms of HIV infection and awareness about HIV transmission routes were frequently reported as advantages of using OSN platforms by Iranian PLWHA. Respondents (n = 27, 93.1%) believed that OSN platforms and online discussions convinced them to use prevention strategies, particularly while communicating with their seropositive as well as seronegative partners.

*Improving knowledge about drug benefits/side effects*. Respondents (n = 25, 86.2%) stated that OSN platforms helped them by giving information about medications used for the HIV cascade. PLWHA discussed the potential side effects of HIV medications and their potential interactions with other medications. Health workers and physicians provided PLWHA with instructions about correct using medications and dosage, complementary therapies, and dealing with side effects of antiretroviral therapies.

*Providing information about available health services and service*. Respondents (n = 8, 27.6%) believed that using OSN platforms assisted Iranian PLWHA to retain local HIV health care services and provide them with information about health centers delivering quality health services to these patients. In addition, linking potential patients and high-risk people to service providers and encouraging them to request an HIV rapid diagnostic test was reported by online community members as benefits of OSN platforms.

**Theme 5: Enhancing perception and attitudes towards HIV.** *Enhancing the perception of users about acquiring HIV infection*. Most respondents (n = 11, 37.9%) believed that using OSN platforms influenced the beliefs and perceptions of OSN users about contracting HIV infection. Due to a lack of HIV-AIDS knowledge, users engaging in high-risk behaviours tend to perceive themselves as less susceptible to HIV/AIDS. By improving awareness and knowledge about HIV/AIDS, OSN platforms served vulnerable users to engage in preventive activities and change their risky behaviours.

*Enhancing the perception of users about negative outcomes and severity of the disease*. According to respondents' perspectives (n = 17, 58.6%), using OSN platforms by Iranian PLWHA was associated with positive perceptions about the severity of symptoms, and treatment outcomes of HIV infection such as pain, disability, death, or negative emotions. Most respondents perceived that HIV was a life-threatening and fatal illness without any treatment. It seemed that PLWHA exposure to positive emotions, experiences, and information could improve the negative attitudes of online community members towards HIV consequences.

*Improving perceptions about potential benefits of adapting prevention behaviours*. It was reported that sharing information about the effectiveness of treatment regimens and preventive interventions among PLWHA through OSN platforms was associated with improved

perception of users. Respondents (n = 5, 17.2%) believed that a positive perception of the benefits of treatments was associated with medication adherence among PLWHA.

*Improving users' knowledge about social, personal, environmental, or economic challenges of living with HIV infection.* Sharing personal experiences about different social, environmental, and economical consequences of living with HIV infection; allowed PLWHA to identify potential barriers that prevented them from changing their risky behaviours or starting ART regimens. In this regard, two respondents believed that OSN platforms helped PLWHA understand what barriers patients encountered in adhering to ART over time, and provided them with effective solutions to increase their medication adherence.

**Theme 6: Social support.** *Facilitating emotional support exchange among users.* Social support was reported as a supportive component of OSN technologies that facilitated coping with HIV/AIDS, improving mood, and expediting recovery from the disease among PLWHA. Social support refers to the perception that one is cared for, valued, and has assistance available from family, friends, neighbours, co-workers, and members of supportive social networks. These supports can be emotional, informational, or instrumental. Emotional support was frequently reported by many participants (n = 17, 58.6%) as the primary reason for using OSN platforms. These participants perceived OSN platforms as sources of empathy, affection, acceptance, trust, encouragement, and caring. It was frequently reported by the respondents that these platforms have made them feel that they were valued and accepted. Almost all respondents mentioned this component during the interviews (n = 53).

*Facilitating instrumental support among users.* It was occasionally reported that OSN platforms had provided some sort of instrumental supports for PLWHA (n = 6). Respondents reported some examples of assisting other members or online friends through networking and group announcements.

**Theme 7: Health-related outcomes.** *Boosting motivation and self-confidence.* According to interviews with PLWHA, using OSN technologies has been associated with some specific psychological outcomes including motivation and self-confidence (n = 22), self-efficacy and self-care (n = 11), and medication adherence (n = 22).

Social interactions with peer groups through OSN technologies have offered the PLWHA the benefit of enhancing their emotional and functional well-being. By observing other similar patients' health status and sharing success stories, PLWHA were motivated and inspired to follow their treatment plan and handle the stress and negative consequences of the disease successfully.

Data analysis revealed that social interactions with other similar patients online encouraged PLWHA to express their personal emotions more frequently. They usually tried to affect other users by sharing inspiring texts, images, and videos and expressing positive emotions. Sharing the good news, funny stories, and jokes were often reported by respondents to improve attitude among PLWHA. Most respondents believed that a positive attitude helped them cope with stress, improve their well-being and even boost their immune system. However, negative emotions were not usually expressed by PLWHA in online groups. Most PLWHA tend to relieve their negative emotions by expressing their fear, anxiety, anger, or sadness by chatting with a close friend or a peer patient privately.

*Improving self-efficacy and self-care behaviours.* The effectiveness of OSN platforms in producing healthier behaviours among PLWHA was reported by some respondents (n = 11). They believed that social interactions with other similar patients and exposure to public health communication encouraged them to change their behaviours and routine lifestyle. Examples include increased physical activity, a healthy diet; smoking cessation, and quitting alcohol consumption. Improving perceived self-efficacy was also reported as another benefit of OSN groups for PLWHA (n = 11). According to findings from qualitative interviews, online

interactions with peer groups have influenced PLWHA beliefs about their capabilities to actively manage their disease and adhere to ART regimens even in the presence of barriers. PLWHA, who felt more self-efficacious and confident during the HIV care continuum, engaged in their treatment plan more actively and reached better outcomes (i.e., better medication adherence or quality of life).

*Improving medication adherence.* Improved adherence to ART regimens emerged as the final supportive component of OSN platforms for PLWHA. Most respondents (n = 22) reported that online discussions about "keeping track of daily medications" were associated with improvements in adherence and medication pick-up. Examples included some patients, who had imperfect or occasional non-adherence to ART during their treatment course and OSN platforms assisted them to improve their medication adherence.

## Phase II results: Delphi study

In this phase, the Delphi study was conducted to examine the consensus of Iranian PLWHA on the proposed conceptual model. The consensus rates of items related to the supportive role of OSN technologies in improving health literacy and medication adherence of PLWHA are summarized in Table 3. According to the results of the Delphi study, all items related to the supportive role of OSN technologies reached the acceptable level of consensus (75% agreement) and were confirmed.

**Table 3. The consensus rates of items related to the role of OSN technologies in improving health literacy and medication adherence of PLWHA.**

| Items | Median | Inter-quartile range | Consensus rate (%) |
|---|---|---|---|
| Improving trust among users | 5 | 1 | 88.4 |
| Facilitating communication between patients and peer groups | 5 | 1 | 96.2 |
| Facilitating communication between patients and health care providers | 5 | 1 | 84.7 |
| Creating identity and a sense of belonging to the group | 5 | 1 | 80.7 |
| Facilitating remote access to information | 5 | 1 | 80.7 |
| Maintaining confidentiality and anonymity of patients in accessing information | 5 | 1 | 92.2 |
| Facilitating access of hidden populations to information | 4 | 1 | 76.8 |
| Promoting interactive Q and A | 5 | 1 | 92.3 |
| Strengthening the motivation of users to share personal experiences | 5 | 1 | 92.3 |
| Facilitating observational learning from the personal experiences of other patients | 5 | 1 | 92.3 |
| Improving patient empowerment and decision making | 5 | 1 | 80.7 |
| Social capital | 4 | 1 | 88.4 |
| Getting familiar with clinical manifestations of the disease | 5 | 1 | 76.9 |
| Getting familiar with the benefits/ side effects of medications | 5 | 1 | 96 |
| Getting familiar with the type of health services and centers providing these services to PLWHA | 5 | 1 | 84.5 |
| Strengthening the sense of vulnerability in patients | 4.5 | 1 | 84.6 |
| Strengthening the patient's attention to serious complications **of** the disease | 5 | 1 | 80.7 |
| Strengthening the patients' understanding of the benefits and effectiveness of treatment advice provided by health professionals | 5 | 1 | 92.2 |
| Strengthening the patient's understanding of barriers to following treatment advice and taking action | 5 | 1 | 80.7 |
| Improving emotional and social support | 5 | 1 | 88.4 |
| Improving instrumental and physical support | 5 | 1 | 76.8 |
| Increasing motivation and self-confidence | 5 | 1 | 88.4 |
| Improving self-efficacy and self-care | 5 | 1 | 88.5 |
| Improving adherence to antiretroviral medications | 5 | 1 | 92.3 |

### Phase III results: The NGT panel

In this phase, the proposed conceptual model was discussed by the NGT panel, and the trust-worthiness of the model was finally approved by the entire panel members (100% agreement). As a result, the final model was illustrated, as shown in Fig 1. Following the panel comments, the causal and contributing factors were also conceptualised between communication, information, and social support; motivation and self-confidence; self-efficacy; and medication adherence among Iranian PLWHA.

## Discussion

The study revealed that OSN technologies supported the HIV care continuum by improving health literacy and medication adherence through several components. By enhancing the social interactions of PLWHA with peer groups and health care providers, OSN platforms offered the patients the opportunity to experience a sense of trust, collective identity, and belonging. By facilitating the access of PLWHA to information and personal experiences of other similar patients, OSN platforms were perceived as attractive venues for seeking/sharing online health information. OSN platforms were also effective at augmenting social support

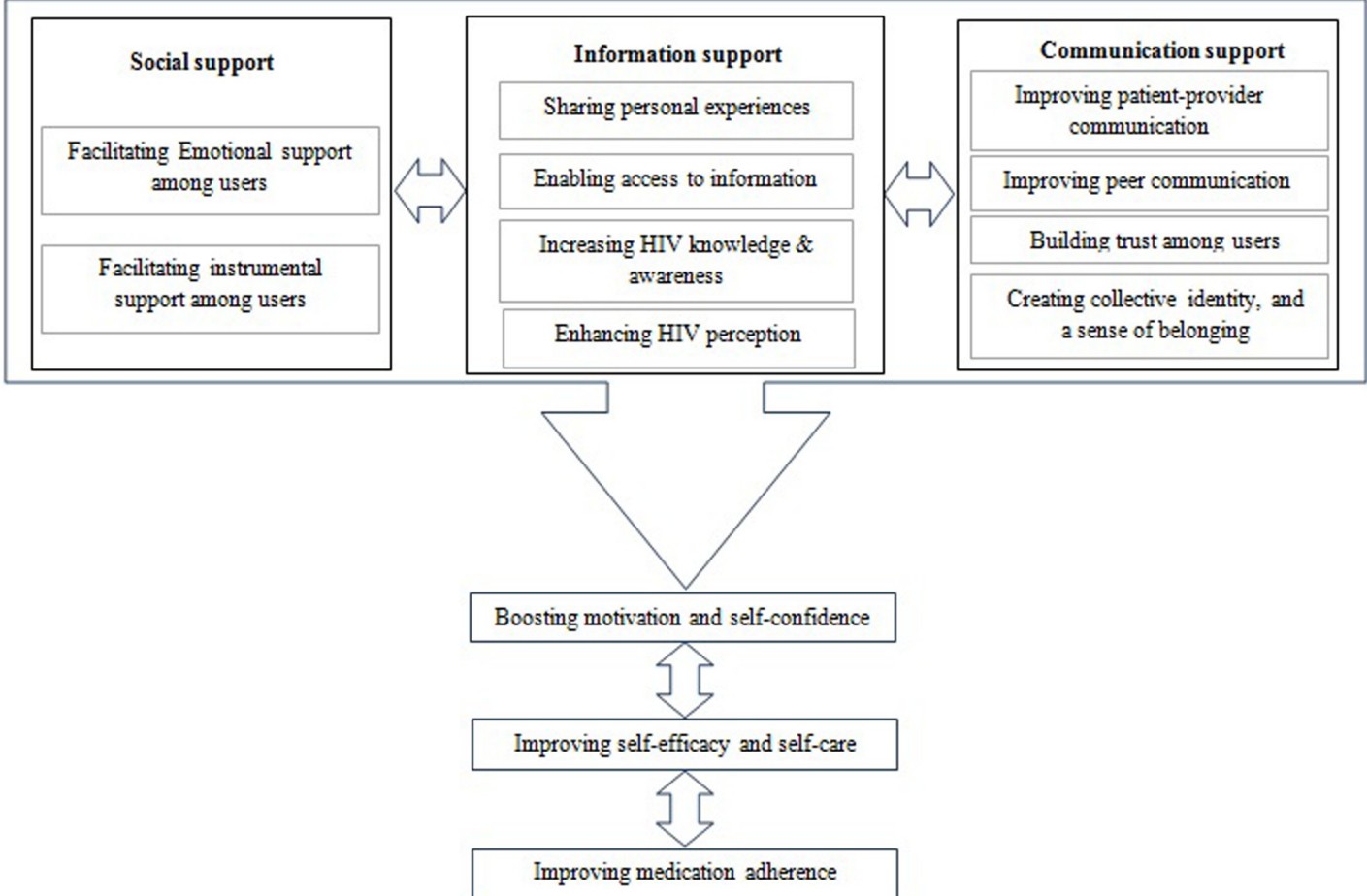

**Fig 1. The proposed model of the roles of online social network technologies in improving health literacy and medication adherence of people living with HIV/AIDS.**

exchange and subsequently improved several health-related outcomes including motivation, self-confidence, self-efficacy, and medication adherence among Iranian PLWHA. The Delphi study obtained consensus from a panel of 26 Iranian PLWHA who participated in the first round of the study, on 100% of the 24 statements proposed to develop the conceptual model of the key role of OSN tools in improving health literacy, and medication adherence among PLWHA. Additionally, the credibility, trustworthiness, and validity of the proposed model were confirmed using an NGT method. Discussions about important constructs and variables that should be considered as critical roles of OSN tools are outlined below under the subheading's health literacy and medication adherence.

## Health literacy

The direct effect of using OSN platforms on health literacy is also investigated and the results suggest the promising role of these platforms in improving the health and health literacy of PLWHA. Consistent with existing evidence, the results suggest that sharing reliable health information by healthcare providers and personal experiences about coping with HIV infection is associated with increased health literacy among patients. The advantages of social media and OSN platform for improving the health literacy of patients with chronic diseases were previously investigated [26, 27] and the results confirmed the effectiveness of OSN interventions in improving the health literacy of patients with cancer [28], diabetes [29–31], and other chronic diseases [32, 33].

According to qualitative interviews and the Delphi study, using OSN tools is associated with increased health literacy through a list of components including communication support. Findings from the Delphi study represented OSN tools as viable options for PLWHA to interact with physicians and care providers. The widespread use of OSN technologies among both physicians and patients has proven to cast a positive impact on the quality of patient-doctor interactions. By increasing their knowledge, patients become motivated to actively communicate with their physicians during their medical consultations. Additionally, OSN technologies can empower patients to follow physicians' recommendations, adhere to proposed treatment plans, and discuss their questions and concerns, especially within an online support group. Despite significant results from existing literature indicating patient benefits [34, 35] empirical research on online patient-doctor communication returns for physicians is lacking. In addition, some evidence fails to indicate the positive perception of physicians and medical students regarding patient-doctor interactions through OSN technologies including Facebook particularly due to lack of privacy and confidentiality issues [35–38]. Further research should investigate how Iranian physicians and care providers feel about patient interactions within OSN technologies and their motivations behind these interactions.

Improving peer communication is another contributing factor that is associated with health literacy among PLWHA. This is stated by around two-thirds of the PLWHA who participated in the qualitative interviews, that OSN technologies had the advantage to communicate with their old friends and make connections with new friends. The results presented are consistent with recent evidence that OSN sites are increasingly referred to as ways of keeping in touch with older friends, forming communications with new friends, and maintaining contact with existing friends and peers [39]. It seems that facilitating peer communication among PLWHA will contribute to improving the health literacy of patients by optimizing the dissemination of HIV prevention messaging and harm reduction strategies [40].

According to our findings, seeking/sharing online health information emerged as one of the key roles of OSN platforms in improving the health literacy of PLWHA. This finding is consistent with preliminary research findings that PLWHA spent more time online using

educational and informational components of OSN apps [12] for seeking [41] and disseminating HIV information [42]. However, disparities in the use of the Internet and OSN tools for finding health information exist. Increased use of OSN technologies for obtaining HIV-related information online has been reported in youth [43], PLWHA with lower ART adherence [14], and people with low health literacy [44], indicating insufficient ability to acquire and act on information related to HIV management. OSN technologies have shown promise in improving PLWHA access to health information. However, for OSN interventions to be effective, they need to be developed and optimized with PLWHA needs and expectations.

The current study findings also suggest that using OSN technologies is associated with increased HIV knowledge among PLWHA. PLWHA use OSN platforms to communicate with their peers about sexual health discuss personal experiences and learn about HIV and sexually transmitted infections prevention. These findings are consistent with prior research findings indicating the effectiveness of using OSN technologies in improving HIV knowledge among PLWHA [39, 45, 46] and other high-risk groups [47] such as homeless youth [48, 49]. OSN platforms are increasingly used by health agencies such as the CDC and aids.gov to inform high-risk populations about the risks of HIV/AIDS, and sexually transmitted infections. In addition, using OSN interventions to reach and inform hidden populations and other vulnerable communities such as female sex workers and injecting drug users demonstrated the advantages of OSN tools for HIV prevention [50–52].

## Medication adherence

Our findings represented the consensus of the Iranian PLWHA on the key role of OSN tools in improving medication adherence and health-related outcomes among patients. One of the most important contributing factors that may be strongly associated with health-related outcomes among PLWHA is social support. Participants generally believed that OSN technologies were effective at augmenting social support exchange by illustrating that informational, emotional, and instrumental supports were exchanged most frequently among PLWHA. This could be partly due to the promising role of OSN apps in addressing a range of PLWHA's individual needs including a need for cognition (information gathering), a need to belong (gaining social approval, expressing opinions, and influencing others), and collective self-esteem [39]. These findings are consistent with other research where it is demonstrated that OSN use brings the advantage of supporting members by providing informational and emotional coping strategies regarding their health concerns [12]. The role of social support on both the physical and psychological well-being of PLWHA has been extensively documented [53–55]. Additionally, online support groups have been demonstrated to have a positive effect on disclosure, mental wellbeing, and engagement with HIV care [56–58]. Social support has also been associated with less depression [59, 60], positive health behaviours, improved ART adherence [61, 62], coping, and quality of life [63, 64], through its functional components including informational, emotional, and instrumental support.

Based on our findings from qualitative interviews, OSN technologies appear to improve medication adherence behaviours among PLWHA. Although there is no absolute cure for HIV infection, consistent and effective use of antiretroviral therapy (ART) can control the infection and prevent transmission of the virus to others. However, poor, or non-adherence to ART regimens is a serious barrier to the successful management of HIV/AIDS worldwide. It can result in disease progression, unsuccessful viral suppression and resistance to medication, opportunistic infections, poor health outcomes, decreased quality of life, and a higher risk of mortality. Unfortunately, poor adherence and treatment interruption have been commonly reported among Iranian patients with HIV/AIDS [1, 65–67]. Several factors have been

evidenced to affect medication adherence in patients with HIV/AIDS including self-efficacy [68], uncertainties about the effects of medications and disease progression, stigma [69], family responsibilities and social support received from family members and friends, problems with schedule and routine [69] and drug abuse [65]. Inadequate functional health literacy has been also proposed as a key driver of poor adherence in patients with HIV/AIDS [70]. According to our findings from qualitative interviews and the Delphi study, using OSN technologies can address the challenge of poor medication adherence among PLWHA by improving some of the drivers of medication adherence including health literacy, social support, and self-efficacy. These findings are consistent with prior studies demonstrating the effectiveness and feasibility of online peer-to-peer social support interventions to improve ART adherence in patients [57, 58, 71, 72]. The findings of the present study should be considered in the context of several limitations. First, our findings are derived from data collected by purposive sampling of PLWHA. As such, the findings are likely not generalizable to the broader population of Iranian PLWHA. The findings are also likely biased by some degree of self-selection, where those who are willing to engage with health facilities and community centers regarding HIV prevention and risk reduction, indicating higher levels of health literacy and perceived susceptibility and severity may have been more likely to participate in the study. According to health belief theory, individuals with higher levels of perceived susceptibility or severity are more likely to engage in health-related behaviours to prevent the health problem or disease from occurring or reduce its consequences [73].

## Conclusion

The study demonstrated the promising roles of OSN platforms in improving health literacy and medication adherence of Iranian PLWHA. OSN tools are globally accessible and acceptable to PLWHA and high-risk subgroups. Therefore, these tools can be leveraged to provide in-demand information and social support for PLWHA across the world. This study has outlined important constructs and variables that should be taken into account as critical roles of OSN tools, which can be used by healthcare providers as a blueprint to maximize efforts in health literacy and medication adherence of PLWHA through online communities. The proposed model also appears to be a valuable aid in guiding health professionals to understand the factors that encourage patients to highly participate in online social networking activities. Such an approach can enhance the effectiveness of OSN-based interventions as prevention and control strategies at both theoretical and practical levels. Given the ongoing struggles to keep up with the fast-moving technology and innovation, a model that explains OSN engagement on improving health literacy as well as health-related outcomes among PLWHA is a crucial tool for the emerging and a growing technological era of mobile health.

## Supporting information

**S1 Appendix. Coded segments for a sample interview in the Farsi language.**
(DOCX)

**S2 Appendix. Thematic analysis codes system.**
(DOCX)

**S1 File.**
(HTM)

**S2 File.**
(PDF)

## Acknowledgments

The authors would like to thank the HIV/AIDS Office, Iranian Ministry of Health for technical support. We acknowledge Dr.Afsar Kazerouni, Nazanin Heidari, Elham Rezaiee, Zahra Bayat Jozani and Nasrin Kordi for conducting the initial correspondences and contacts with patients. We acknowledge participants for the time and insights they offered during the study.

## Author Contributions

**Conceptualization:** Azam Bazrafshani, Sirous Panahi, Hamid Sharifi, Effat Merghati-Khoei.

**Data curation:** Azam Bazrafshani.

**Formal analysis:** Azam Bazrafshani.

**Methodology:** Sirous Panahi, Hamid Sharifi.

**Project administration:** Sirous Panahi.

**Supervision:** Hamid Sharifi, Effat Merghati-Khoei.

**Writing – original draft:** Azam Bazrafshani, Effat Merghati-Khoei.

**Writing – review & editing:** Azam Bazrafshani, Sirous Panahi, Hamid Sharifi, Effat Merghati-Khoei.

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
