## [Decision Letter · Decision Letter 0]

2 Jun 2021

PONE-D-21-04415

The role of online social networks in improving health literacy and medication adherence among people living with HIV/AIDS in Iran: A mixed-method study

PLOS ONE

Dear Dr Sirous Panahi,

Thank you for submitting your manuscript to PLOS ONE. After careful consideration, we feel that it has merit but does not fully meet PLOS ONE’s publication criteria as it currently stands. Therefore, we invite you to submit a revised version of the manuscript that addresses the points raised during the review process.

We look forward to receiving your revised manuscript.

Kind regards,

Syed Imran Ahmed, MClinPharm, PhD, MRSPH

Academic Editor

PLOS ONE

Journal Requirements:

2. Please include additional information regarding the questionnaire and interview guide used in the study and ensure that you have provided sufficient details that others could replicate the analyses. Specifically, please include a copy of your surveys/interview guides, in both the original language and English, as Supporting Information.

3.We note that you have indicated that data from this study are available upon request. PLOS only allows data to be available upon request if there are legal or ethical restrictions on sharing data publicly. For information on unacceptable data access restrictions, please see http://journals.plos.org/plosone/s/data-availability#loc-unacceptable-data-access-restrictions.

Additional Editor Comments:

Thank you for your submission for consideration to be be published. Due to COVID19 delays it was dfificult to get suitable reviewers to get review reports in time, which has caused delays in getting back to you with any decision. Thank you for your patience.

The reviewer has highlighted comments that are needed to be addressed before the manuscript can be considered. The main comments are on methodlody and conceptual framework of analysis, which i believe need to be very robust as commented by the reviewer. I invite you to kindly take note of all the comments and revised manuscript accoridngly. While doing this revision, kindly ensure that authors instructions are carefully adhered.

Thank you.

Reviewers' comments:

Reviewer's Responses to Questions

**Comments to the Author**

1. Is the manuscript technically sound, and do the data support the conclusions?

Reviewer #1: Yes

2. Has the statistical analysis been performed appropriately and rigorously? 

Reviewer #1: Yes

3. Have the authors made all data underlying the findings in their manuscript fully available?

Reviewer #1: Yes

4. Is the manuscript presented in an intelligible fashion and written in standard English?

Reviewer #1: Yes

5. Review Comments to the Author

Reviewer #1: Please see attached.

6. PLOS authors have the option to publish the peer review history of their article (what does this mean?). If published, this will include your full peer review and any attached files.

Reviewer #1: No

---

## [Author Response · Author response to Decision Letter 0]

25 Jun 2021

Response to reviewers

07 May 2021 

Dear Editor,

Re: PONE-D-21-04415: The role of online social networks in improving health literacy and medication adherence among people living with HIV/AIDS in Iran: Development of a conceptual model

Thank you for giving us the opportunity to submit a revised draft of the manuscript for publication in the Journal of PLOS ONE. We appreciate the time and effort that you and the reviewers dedicated to providing feedback on our manuscript and are grateful for the insightful comments on and valuable improvements to our paper. We have incorporated most of the suggestions made by the reviewers. Those changes are highlighted within the manuscript. Please see below, in blue, for a point-by-point response to the reviewers’ comments and concerns. All page numbers refer to the revised manuscript file with tracked changes.

Reviewer comments to the authors

General

• The manuscript is generally well-written. However, it contains a few grammatical errors and need for syntax improvement. The paper will benefit from careful proof-reading before it will be considered for publication in the journal. 

Thank you! We have revised the grammar and writing of the manuscript again to improve the language and the quality of writing. 

• My major concern with the study is that it is unclear if the Delphi and the nominal group technique were robust enough to yield a valid conceptual model. More information may be required. 

Thank you. We agree with the reviewer. Further information about the technical aspects of the Delphi study and the nominal group technique were added to the method’s section. 

• Another concern is that there is no sufficient information on the items that were used in the questionnaire utilized in the Delphi phase. 

Thank you. We agree with the reviewer. Further information about designing the questionnaire and survey items used in the Delphi study were added to the method’s section. 

• As a normal scientific writing convention, abbreviations should always be defined in full-term at first use and the abbreviation used thereafter. 

We agree with the reviewer’s assessment. Accordingly, throughout the manuscript, we have revised the abbreviations and corrected the typos.

Title

• I am not sure, but I doubt if the study can be described as a “mixed-methods”. In the secondary title, can it be described as “a multi-methods study” or simply “development of a conceptual model”?

The title was edited as you pointed out. 

Abstract

• The abstract is well-written.

• Abbreviations such as HIV should be defined in full-term at first use. 

We agree with the reviewer’s assessment. Accordingly, throughout the manuscript, we have revised the abbreviations and corrected the typos

• While the study is described as three phases, I do not seem to see the third phase in the abstract. I understand the qualitative part as one phase and Delphi as another phase. What is the third phase? 

We have considered the nominal group technique as the third phase of the study.

• Line 32: Change “24 sub-themes were emerged” to “24 sub-themes emerged”. 

Modifications were made as you pointed out.

Introduction

• The Introduction is fairly well-written, and has provided some insight about what is known in the literature and the gaps, but it can be better. 

Some explanations were added to the introduction section as you pointed out.

• However, some redundancies and repetitions regarding the use and value of Internet and OSN among PLWHA. This can be better by minimizing the redundancies.

Modifications were made as you pointed out.

• As a normal scientific writing convention, abbreviations should always be defined in full-term at first use and the abbreviation used thereafter. Examples HIV/AIDS, ART, PLWHA etc. 

We agree with the reviewer’s assessment. Accordingly, throughout the manuscript, we have revised the abbreviations and corrected the typos

• Line 58-61: Any references to support the statements in line 58 - 61? 

New references were added to the manuscript.

• Line 62: The open bracket before OSN should be closed. 

Modifications were made as you pointed out.

• Line 74: A cross-sectional” instead of “Across”. 

Modifications were made as you pointed out.

• Line 81: “patient-cantered” should be “patient-centered” 

Modifications were made as you pointed out.

• Aim of the study: The aim in the Introduction is different from the one in the Abstract. The researchers need to be consistent. 

Modifications were made as you pointed out.

Methods

• Line 98-99: Summarize the three sequential phases and justify the choice of this approach. It also must be well clear what you aim to achieve in phase 2 and phase 3 and how they are connected. 

Some explanations were added to the method section as you pointed out.

• Line 122: You said interviews duration ranged between 14 - 60 minutes and the longest lasted for 90 minutes. This is a contradiction. 

Modifications were made as you pointed out.

• Line 123: Change to “… were conducted”. 

Modifications were made as you pointed out.

Phase 2 and Phase 3: The process for the development of conceptual model is rather superficial and not detailed enough. More information and robust process are needed. Some explanations about the techniqual aspects of the Delphi study and development of the conceptual model were added as you pointed out.

• Phase 2 (Delphi study): It is unclear how many rounds of Delphi were conducted. 

• Modifications were made as you pointed out.

• Phase 3 (Nominal group technique): It is unclear how the NGT differs from the Delphi in terms of methodology. I am also not clear how many members were included in total. If n=5, is this considered adequate?

• Modifications were made as you pointed out.

• Line 141: Was online social networks not abbreviated as OSN before? 

• Modifications were made as you pointed out.

• Line 148 and throughout the paper: British vs. US English.

• Modifications were made as you pointed out.

Results

• Generally well-written with clear tables to support the textual descriptions. 

Thank you!

• Line 171 – 183: Demographic information should have also been presented in a table. A new table was inserted to the manuscript to summarize the demographic data of respondents.

• Line 182: Thirteen of 29 does not represent 51%.

• Modifications were made as you pointed out

• Line 192: Change “retroviral” to “anti-retroviral” 

• Modifications were made as you pointed out

• Line 192: Are you sure there are four sub-themes? I counted 5 in Table 1. 

• Modifications were made as you pointed out

• Line 411: In the Delphi methods, the level of agreement was 75%. Here you have changed to 70%. Be consistent. 

Modifications were made as you pointed out

Discussion

• The Discussion needs to be more critical and supported by relevant literature and not merely a summary of the results. 

• The discussions are redundant in places. These must focus on the major findings and citing relevant literature. Several important previous studies not cited. 

Thank you! We have modified the discussion and classified the reference to a new format as you pointed out. 

Conclusion

• Can be better written to reflect the study objectives and finding. In addition, how would the conceptual model help in designing interventions to improve medication literacy and medication adherence through OSN? 

Conclusion was revised and implications of the proposed model were added. 

References

• I advise that the authors should strictly adhere to the journal’s referencing style (please see authors instructions). 

• Some references are incomplete and inaccurate.

• In addition, some journals names are not according to the journal’s style.

Thank you! All journal names were checked by the NLM journal database and the abbreviations were added. Reference styles were again checked by the Vancouver style.

---

## [Decision Letter · Decision Letter 1]

16 Sep 2021

PONE-D-21-04415R1The role of online social networks in improving health literacy and medication adherence among people living with HIV/AIDS in Iran: Development of a conceptual modelPLOS ONE

Dear Dr. Panahi,

Thank you for submitting your manuscript to PLOS ONE. After careful consideration, we feel that it has merit but does not fully meet PLOS ONE’s publication criteria as it currently stands. Therefore, we invite you to submit a revised version of the manuscript that addresses the points raised during the review process.

Thank you for your revised submission. The reviewers are generally pleased with the efforts, however, recommended some changes including language editing. Please ensure the mansucript proof reading before revised submission for quality word processing.

We look forward to receiving your revised manuscript.

Kind regards,

Syed Imran Ahmed, MClinPharm, PhD, MRSPH

Academic Editor

PLOS ONE

Journal Requirements:

Reviewers' comments:

Reviewer's Responses to Questions

**Comments to the Author**

1. If the authors have adequately addressed your comments raised in a previous round of review and you feel that this manuscript is now acceptable for publication, you may indicate that here to bypass the “Comments to the Author” section, enter your conflict of interest statement in the “Confidential to Editor” section, and submit your "Accept" recommendation.

Reviewer #1: All comments have been addressed

Reviewer #2: All comments have been addressed

2. Is the manuscript technically sound, and do the data support the conclusions?

Reviewer #1: Yes

Reviewer #2: Yes

3. Has the statistical analysis been performed appropriately and rigorously? 

Reviewer #1: N/A

Reviewer #2: I Don't Know

4. Have the authors made all data underlying the findings in their manuscript fully available?

Reviewer #1: Yes

Reviewer #2: Yes

5. Is the manuscript presented in an intelligible fashion and written in standard English?

Reviewer #1: No

Reviewer #2: Yes

6. Review Comments to the Author

Reviewer #1: Thank you for addressing most of my previous comments. I have just 2 minor comments for your consideration:

1. I still believe the paper need language editing. There are still minor grammatical errors, needs for syntax improvement, and other issues (e.g. starting a sentence with small letters, using symbols such as & in sentences etc).

2. I believe Table 2 containing the qualitative results can be better formatted. For example, I am not sure if each quote example aligns well with the relevant sub-theme.

Reviewer #2: The language of the manuscript should be cross-checked and modified, accordingly. It requires improvement. If it could be copy edited by a professional copy-editor, it would be nice.

7. PLOS authors have the option to publish the peer review history of their article (what does this mean?). If published, this will include your full peer review and any attached files.

Reviewer #1: No

Reviewer #2: No

---

## [Author Response · Author response to Decision Letter 1]

12 Oct 2021

Dear reviewers,

Thank you so much for all your insightful comments. Our manuscript’s language has been recently evaluated by an expert English Language researcher and we have made some modifications to improve the quality of the grammar and writing style of the manuscript. We have also checked the reference style. In addition, we have tried to improve the format of the qualitative results in Table 2. The co-authors have checked the open codes and sub-themes carefully. But they have decided not to change the current format. We really appreciate your dedication and commitment to improve the quality of this manuscript. Hope these modifications comply with your remarks. 

Sincerely yours,

Sirous Panahi

Corresponding author

---

## [Decision Letter · Decision Letter 2]

1 Dec 2021

The role of online social networks in improving health literacy and medication adherence among people living with HIV/AIDS in Iran: Development of a conceptual model

PONE-D-21-04415R2

Dear Dr. Sirous Panahi,

We’re pleased to inform you that your manuscript has been judged scientifically suitable for publication and will be formally accepted for publication once it meets all outstanding technical requirements.

Kind regards,

Syed Imran Ahmed, MClinPharm, PhD, MRSPH

Academic Editor

PLOS ONE

Additional Editor Comments (optional):

Reviewers' comments:

Reviewer's Responses to Questions

**Comments to the Author**

1. If the authors have adequately addressed your comments raised in a previous round of review and you feel that this manuscript is now acceptable for publication, you may indicate that here to bypass the “Comments to the Author” section, enter your conflict of interest statement in the “Confidential to Editor” section, and submit your "Accept" recommendation.

Reviewer #1: All comments have been addressed

2. Is the manuscript technically sound, and do the data support the conclusions?

Reviewer #1: Yes

3. Has the statistical analysis been performed appropriately and rigorously? 

Reviewer #1: Yes

4. Have the authors made all data underlying the findings in their manuscript fully available?

Reviewer #1: Yes

5. Is the manuscript presented in an intelligible fashion and written in standard English?

Reviewer #1: Yes

6. Review Comments to the Author

Reviewer #1: I have reviewed this revised version of this manuscript. The authors have addressed all my concerns. Thank you.

7. PLOS authors have the option to publish the peer review history of their article (what does this mean?). If published, this will include your full peer review and any attached files.

Reviewer #1: No

---

## [Editor Report · Acceptance letter]

20 Jun 2022

PONE-D-21-04415R2 

The role of online social networks in improving health literacy and medication adherence among people living with HIV/AIDS in Iran: Development of a conceptual model 

Dear Dr. Panahi:

I'm pleased to inform you that your manuscript has been deemed suitable for publication in PLOS ONE. Congratulations! Your manuscript is now with our production department. 

Kind regards, 

on behalf of

Dr. Syed Imran Ahmed 

Academic Editor

PLOS ONE